# The Effect of Production and Post-Harvest Processing Practices on Quality Attributes in *Centella asiatica* (L.) Urban—A Review

Rambir Singh [1,*], Balasiewdor Kharsyntiew [1], Poonam Sharma [2], Uttam Kumar Sahoo [3,*], Prakash Kumar Sarangi [4], Piotr Prus [5] and Florin Imbrea [6,*]

1   Department of Horticulture, Aromatic and Medicinal Plants, Mizoram University, Aizawl 796004, India; balakharsyntiew4@gmail.com
2   Department of Zoology, Indira Gandhi National Tribal University, Amarkantak 484886, India; pnm245@yahoo.com
3   Department of Forestry, Mizoram University, Aizawl 796004, India
4   College of Agriculture, Central University of Agriculture, Imphal 795004, India; sarangi77@yahoo.co.in
5   Department of Agronomy, Faculty of Agriculture and Biotechnology, Bydgoszcz University of Science and Technology, Al. Prof. S. Kaliskiego 7, 85-796 Bydgoszcz, Poland; piotr.prus@pbs.edu.pl
6   Faculty of Agriculture, University of Life Sciences "King Mihai I", 300645 Timisoara, Romania
*   Correspondence: sehrawat_r@yahoo.com (R.S.); uttams64@gmail.com (U.K.S.); florin_imbrea@usvt.ro (F.I.)

**Abstract:** *Centella asiatica* is well known for its miraculous therapeutic properties in various systems of traditional medicine across the world. However, significant variation in its pharmacological activities has been reported due to the unavailability of quality raw material and non-standardized formulations. A number of research papers have been published on the collection of *C. asiatica* plants from different regions for the identification of a suitable agroclimate with elite germplasms. Efforts have been made to standardize production and post-harvest practices for the availability of quality raw material with a high centelloside content. The ecological niche modeling approach revealed that the Indian subcontinent has high climatic suitability for the production of *C. asiatica*, and genotypes with a high content of centelloside were predominantly found in the Western Ghats, North East, Eastern Himalaya and Western Himalaya in India. Open cultivation of *C. asiatica* is more suitable in these agroclimatic zones in India. Cultivation under shade is also suitable in the plains of Central India. Hydroponic and tissue culture of *C. asiatica* has also been successfully established for the enhanced production of centelloside using supplements and elicitors such as sucrose, auxins, cytokinins, kinetin, methyl jasmonate, etc. Freeze drying has been identified as the most efficient post-harvest method for the high pharmacological activities of *C. asiatica* extracts.

**Keywords:** *Centella asiatica*; biomass; drying; madecassoside; asiaticoside; madecassic acid; asiatic acid

## 1. Introduction

*Centella asiatica* (L.) Urban (family Apiaceae) is commonly known as Asiatic pennywort, India pennywort and "*Gotu Kola*". *C. asiatica* is used in South East Asia as a traditional medicine for the management of a number of disorders since time immemorial. It has been used widely in the Indian System of Medicine as "brain food" and for brain wellness, improving memory and the management of nervine disorders like epilepsy, schizophrenia and cognitive dysfunction [1]. The plant also possesses wound healing, anti-diabetic, hepatoprotective, antinociceptive and anti-inflammatory properties [2–4]. Owing to its widespread pharmacological activities, significant interest has been generated in this plant. Widespread variations in the pharmacological activities of this plant have been reported due to agroclimate-specific variations in the phytochemicals of collected chemotypes, cultivation practices and post-harvest processing. Cultivation practices leading to high biomass production play a significant role in the commercial viability of a crop. Post-harvest processing is crucial for the quality and quantity of major bioactive compounds responsible

for pharmacological activities. A suitable production system with high biomass output and optimized post-harvest processing is the key to the gainful cultivation of *C. asiatica*.

This plant grows widely in damp and marshy areas during the rainy season, especially in tropical, subtropical and temperate zones. The plant grows up to 700 m above sea level in swampy, moist and marshy environments during the rainy season. The plant is found in the wild in South East Asia, Africa, Australia, the southern part of the United States of America and South America [5]. In India, the plant is known by various vernacular names in different regions and languages, viz. *Chokiora* (Bihar), *Thankuni* (Bengali), *Barmi* (Gujarati), *Mandookaparni* (Hindi), *Supriya*, *Mutthil* (Sanskrit), *Brahmi* (Urdu), *Babassa*, *Saraswataku* (Telegu), *Bat syiar/Khlieng syiar* (Khasi, Meghalaya), *Peruk* (Manipuri), *Lambak* (Mizoram) and *Manimun* (Assam) [1]. *C. asiatica* grows abundantly in moist, sandy or clayey soils, often in large clumps forming a dense green carpet or as a weed in crop fields and other waste places throughout India [1]. It may be cultivated in drier soils as long as they are watered regularly.

The present review summarizes important ethnomedicinal uses, applications in modern medicine, major phytochemicals and agroclimatic specific variations in the phytochemicals of *C. asiatica*. Information is also compiled on various production systems, viz. open and shade, hydroponics and tissue culture and post-harvest processing methods, for the identification of suitable conditions for the production and availability of quality raw material for the growing phytopharmaceutical industry.

## 2. Data Collection

A systematic review of available articles from 2000–2022 was carried out using open-source databases PubMed and Google Scholar. The search keywords were "pharmacological activity of *Centella asiatica*", "Phytochemicals of *Centella asiatica*", "cultivation of *Centella asiatica*", "production of *Centella asiatica*", "processing of *Centella asiatica*" and "tissue culture of *Centella asiatica*". The web source www.sciencedirect.com was also used for retrieving additional literature using these keywords. Articles written in other than the English language, conference papers and proceedings were not included in the review paper. Papers lacking a minimum of 04 types of vital information such as the collection of germplasm, production technology, plant part used for processing, post-harvest processing method, yield parameter and content of centellosides (pooled or individual) were also excluded. The process of the selection of articles for inclusion and exclusion is given in Figure 1. Information regarding the collection site of wild accession for cultivation, collection of the released variety for cultivation, production technology (open cultivation/shade cultivation/organic production system/good agricultural production system/hydroponic/tissue culture production system), plant part used (whole plant/leaves/roots/callus/explant), post-harvest processing (sun drying/shade drying/hot-air oven drying/freeze drying), yield parameters (fresh biomass/dry matter yield) and the content of major centellosides (asiatic acid, madecassic acid, asiaticoside and madecassoside) was tabulated for the easy understanding and ready reference of researchers. The chemical structures of bioactive compounds were drawn by using Chemistry 4D Draw (Chemdraw).

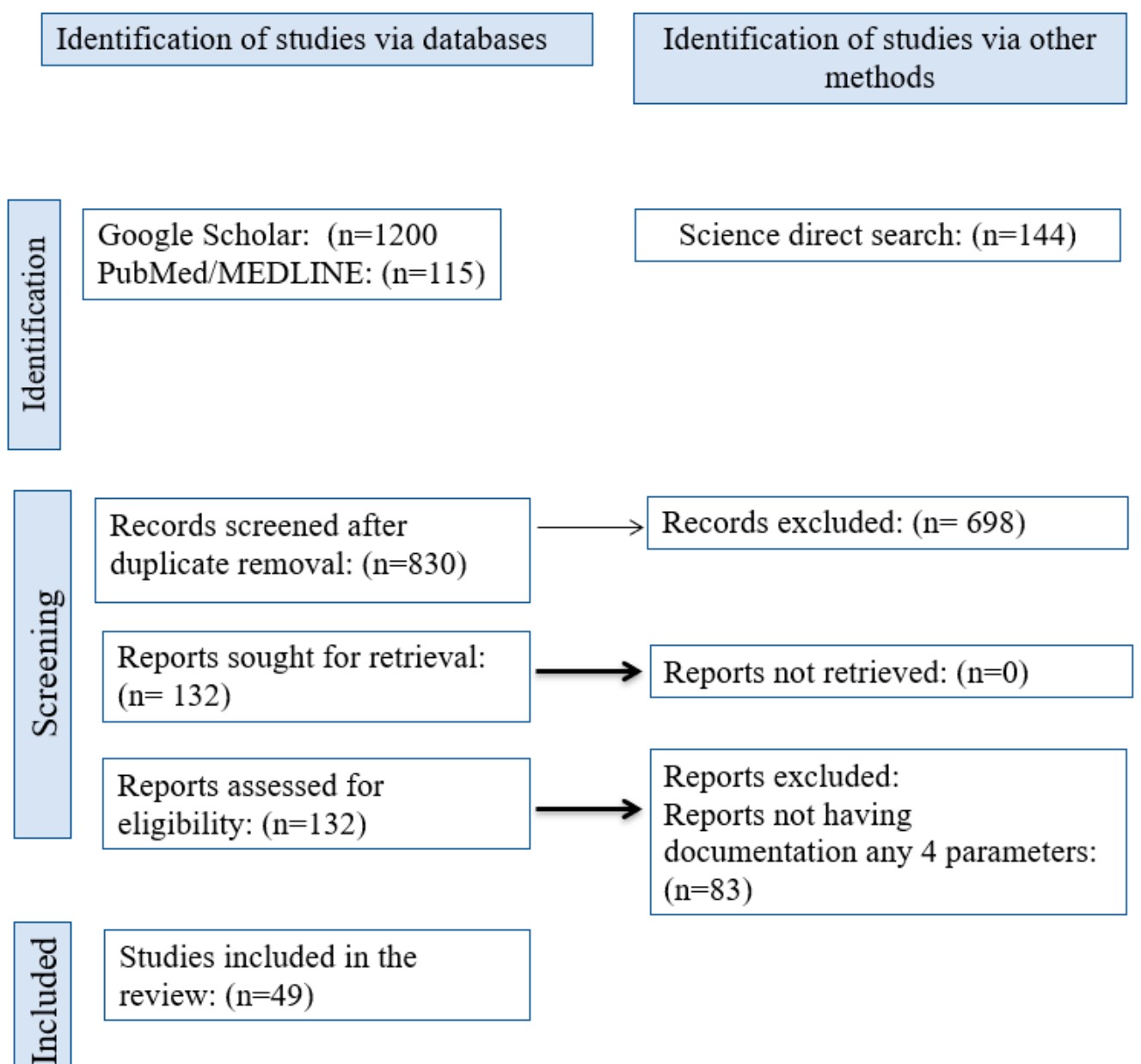

**Figure 1.** Flow diagram of study selection.

## 3. Results and Discussion

The therapeutic properties of *Centella asiatica* have been reported in Ayurvedic medicine in India about 3000 years ago and traditional Chinese medicine about 2000 years ago [6]. The large number of pharmacological activities associated with the plant generated significant interest in the identification of agroclimate-specific elite chemotypes with high centelloside contents as well as production and post-harvest processing technologies for generating quality raw material for commercial cultivation and drug preparation. The plant is abundantly available in South East Asia, particularly India. The identification of the agroclimate with a high abundance of *C. asiatica* and centelloside contents has been attempted by various researchers.

### 3.1. Ethnomedicinal Use of Centella asiatica

The literature survey showed that the plant has a long history of ethnomedicinal use in Asian countries [7,8]. A large number of ethnomedicinal reports are from South East Asia, especially the Indian subcontinent including India, Pakistan, Bangladesh, Sri

Lanka, Nepal, Bhutan and Myanmar. Apart from the Indian subcontinent, its traditional uses have been reported in other Asian countries like China, Japan, Cambodia, Indonesia, Malaysia, Thailand and Vietnam. The ethnomedicinal uses of *C. asiatica* have been reported in ancient medicinal texts such as Indian ayurvedic pharmacopeia [9,10], Chinese herbology medicine [11] and Japanese Kampo medicine [12,13]. There are very few reports of the use of this plant as a fresh vegetable from other parts of the world as well [14]. Ethnomedicinally, the healing properties of *C. asiatica* are vast and vary with ancient cultures and tribal groups [15]. The plant is well known for the treatment of various mild and chronic diseases such as neurological disorders, diabetes, hepatitis, anemia, skin diseases [16], diarrhea, ulcers, fever and amenorrhea [17]. The plant has been widely used as an antioxidant, antibacterial, antiviral and anti-cancer agent [18,19].

The highest number of ethnomedicinal uses of *C. asiatica* has been reported in various ethnic communities in India and also in the ancient texts of the Indian System of Medicine (ISM). The plant has been mentioned as "*Medhya Rasayana*" or *Brahma Rasayana* for the management of mental exhaustion, nervous weakness, insomnia and memory loss and the improvement of overall mental health in the ISM [20]. It has also been used in the Ayurveda and Unani systems for the treatment of ailments like body aches, ulcers, stomach disorders, asthma, leprosy, leukorrhea, urethritis, loose bowels, dysentery, and mental illness [21–23]. There are several reports of the use of *C. asiatica* as "food medicine" for the management of various disorders by ethnic communities in India. Fresh leaves of this plant are consumed as a salad for the management of gastrointestinal disorders like constipation by the ethnic communities of the district of Bandipore, Kashmir, India [24]. Topical application of the fresh leaf paste of this plant is used by ethnic communities in Kolli Hills and Kani Tribals, Tamil Nadu, India, to treat fever and cold [25,26]. In India, Mizo communities in the western part of Mizoram used its leaf decoction for the treatment of asthma and eye problems [27]. The tribes of the Similipal forest, Mayurbhanj, Odisha, India, used one teaspoonful of the leaf paste daily for the treatment of diarrhea and dysentery [28]. The fresh whole plant is used as a cure for stomach problems and for boosting immunity and mental health by the Aka, Miji and Monpa communities of the West Kameng district, Arunachal Pradesh India [29,30]. Various ethnic communities in Kerala, India, used this plant for the treatment of hair diseases, diabetic ulcers, piles, jaundice, anemia, skin diseases and mental illness [31–35].

The plant has been used very profusely in other countries of the Indian subcontinent. In an ethnomedicinal study reported in the district of Rajshahi, Bangladesh, 125 g of the leaves was boiled with water, and 1 cup of the decoction was taken with honey every morning and evening to treat blood disorders. For the management of fever, 31.25 g of the leaf juice of *C. asiatica* was mixed with 31.25 g of the leaf juice of *Nyctanthes arbor-tristis* and taken every morning on an empty stomach [36]. The ethnic communities in Jessore district in Khulna Division, Bangladesh, indicated that the application of *C. asiatica* fresh leaf paste twice daily for 7 days around the nipple resulted in an increase in breast milk after childbirth [37]. The "Garos" community in Madhupur, Tangail, Bangladesh, used whole-plant juice to stop excess menstruation and also whole-plant paste for the treatment of skin diseases [38]. Tripura traditional healers of the Chittagong Hill Tracts region of Bangladesh used *C. asiatica* for gastric disorders, stomach pain and the treatment of diarrhea [39]. Paste prepared from *C. asiatica* young leaves was used in the treatment of eczema and headache, and whole plants were used as a vegetable for the management of dysentery by locals in the village of Dohanagar, Patnitala Upazilla of Naogaon district, Bangladesh [40]. Kavirajes, traditional medicinal practitioners from the Chalna area of Bangladesh, use the whole plant for the treatment of dog bites, asthma, itching, leucorrhea, malaria, tumors and wounds and as a carminative [41].

In Gosiling *gewog* in Tsirang district in the southern region of Bhutan, the ethnic communities reported that the oral administration of the paste of fresh *C. asiatica* leaves helped as a treatment for pneumonia and internal wounds [42]. The plant is used both as a leafy vegetable and a medicinal herb in Myanmar. It has been used for the treatment of

diabetes, skin diseases (eczema, leprosy, itching, rashes and sores), mental illness, blood problems, dysentery, urine retention, painful urination, blood in the urine and syphilitic infections. It has shown snake-poison-neutralizing capacity. The leaf extract, together with sugar and honey, has been given daily to treat colds and fever as a restorative product [43]. In Nepal, tea prepared from *C. asiatica* leaves is used as a detoxicant and diuretic and also for the treatment of diarrhea, hypertension, urinary tract infections and poor memory [44]. The leaf and whole-plant juice has been used as an antiseptic for skin infections [23]. Around four teaspoons of the leaf juice obtained by squeezing 50 leaves are taken orally in the morning for 2–3 weeks to cool the body and stomach [45].

Among other Southeast Asian countries, *C. asiatica* is used as a vegetable and traditional medicine in Malaysia, Indonesia and Thailand [14]. In Malaysia and Indonesia, the whole plant is eaten fresh as a vegetable in a salad, in soup or as an appetizer and is also believed to treat memory loss and act as an anti-aging herbal medicine [46,47]. The Kadazandusun community of Malaysia cook this plant with coconut milk or shredded coconut to prepare soup. In these countries, the plant has been used as tea for treating hypertension, diarrhea and urinary tract infections, as a detoxicant and diuretic and to lower blood pressure and decrease heart rate [23]. In Thailand, the fresh leaves of *C. asiatica* have been eaten with sour chopped meat salad or fried noodles [48]. It has been reported that the freshly prepared juice from the young and tender leaves of *C. asiatica* is a rich source of vitamin A and is commonly taken for thirst-quenching purposes or as a cooling drink to reduce "inner heat" [24].

In traditional Chinese medicine, the plant has been reported for the treatment of the looseness of the bowels, jaundice and scabies, Hansen's ailment (disease), urinary troubles, nosebleeds, breaks, tonsillitis, measles, tuberculosis and other diseases [49]. The dried form of the drug known as *ji xue cao* (at a dosage of 15–30 g) or fresh plant (at a dosage of 30–60 g) has also been prescribed for the treatment of various disorders [50]. *C. asiatica* is also used in TCM to combat physical and mental exhaustion [51].

In Western medicine, it is popular as a "brain tonic", an agent to revitalize the brain and nervous system, increase attention and concentration and combat aging [52,53]. It has been used as a remedy to restore memory and longevity, and when combined with other herbs, it is considered a remedy to retard the symptoms of age and to prevent dementia [54]. Although not native to North America, *C. asiatica* is described as an American Indian remedy (in addition to several other plants including *Ginkgo biloba*, also not native to America) to improve memory [55].

### 3.2. Therapeutic Uses in Modern Medicine

*C. asiatica* has been used as a remedy for several health problems in modern medicine [5]. A summary of the beneficial health effects of this plant is given in Figure 2. The plant has been useful in improving cognitive function and memory [56]. It has shown its effectiveness in the management of mild cognitive impairment (MCI) among the elderly, aged 65 years and above [57]. Administration of *C. asiatica* has protective effects against colchicine-induced cognitive impairment and associated oxidative damage as well as anticonvulsant, antioxidant and CNS-protecting activity in animal models [58]. *C. asiatica* extract can have an influence on neuronal morphology and promote higher brain function during the postnatal developmental stage in mice [56]. The fresh leaf extract could potentially be used to improve neuronal dendrites in conditions involving stress, neurodegenerative disorders and memory disorders [59]. Triterpenes from this plant have shown antidepressant effects in various studies. Triterpenes significantly decreased the serum level of corticosterone and increased the amount of monoamine neurotransmitters in the rat brain [2]. It has been indicated that the constituents/active principles of the plant stimulate neuronal growth, improve memory and prevent neurodegeneration [60]. Asiatic acid isolated from this plant enhanced memory and learning in male Sprague–Dawley rats [61]. This chemical has the ability to reverse the effects of valproic acid treatment on cell proliferation and spatial working memory [62,63].

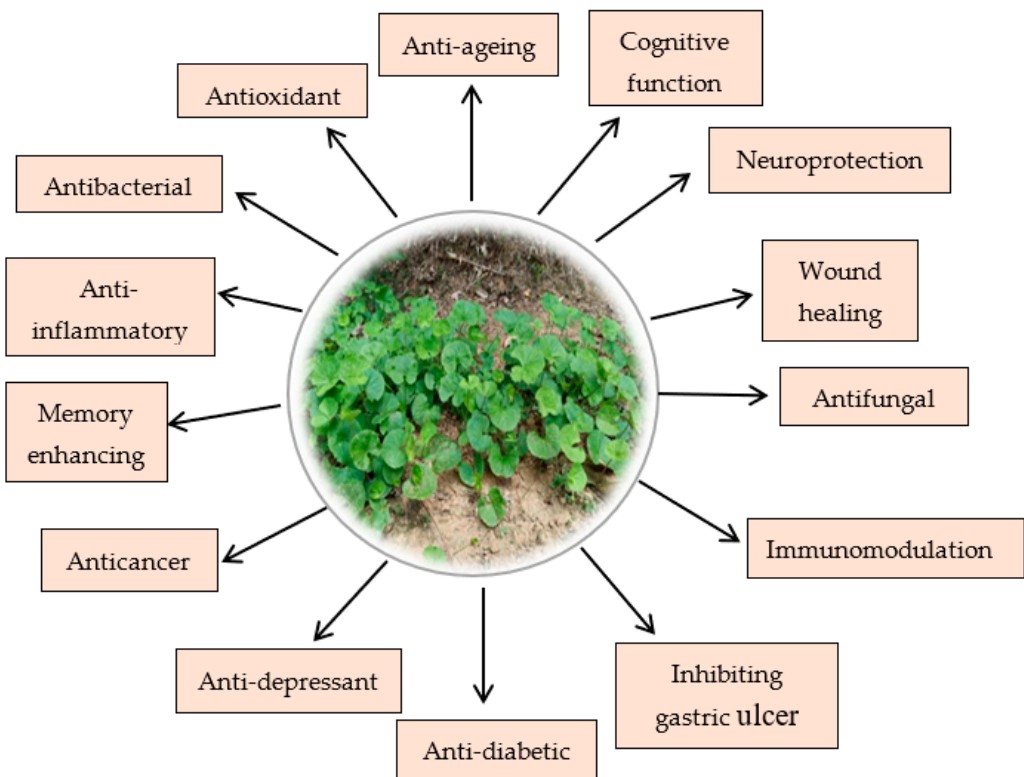

**Figure 2.** Therapeutic uses of *C. asiatica*.

Triterpenoid saponins and pectin found in *C. asiatica* have been shown to exhibit immunomodulatory and immunostimulatory activities, respectively [64,65]. *C. asiatica* extract has been reported to have protective effects against ethanol-induced gastric mucosal lesions [66]. The ulcer-protective effect of the fresh juice is believed to be attributed to its ability to strengthen mucosal defensive factors [64]. Two phytoconstituents, asiatic acid and madecassic acid, possess significant anti-inflammatory activities [67].

Ethanolic and methanolic extracts of *C. asiatica* have demonstrated significant protective effects and the ability to lower blood glucose levels to normal in glucose tolerance tests conducted in alloxan-induced diabetic rats [68].

Asiaticoside has been associated with enhanced wound-healing activity, primarily attributed to increased collagen formation and angiogenesis. It exhibited significant wound-healing activity in animal models of normal and delayed healing [69]. It is also used for the treatment of scars or wounds by increasing the activity of myofibroblasts and immature collagen. It is also used in porcine skin as it stimulates the epidermis by activating the cells of malphigian. Direct application of asiaticoside is also used to treat wounds in rats and increase the ability or strength of newly formed skin. It is used in the treatment of hypertrophic scars and keloids and to prevent new scar formation [1].

Asiatic acid shows cytotoxic effects on human ovarian cancer cells [70]. The antioxidant activity of the plant extract prevented ethanol-induced gastric mucosal lesions by strengthening the mucosal barrier [69]. The plant essential oil, produced by steam distillation, was found to be an excellent antioxidant. Its antioxidant activity was comparable to that of the synthetic antioxidant butyl hydroxy anisole [71].

Two new flavonoids, castilliferol and castillicetin, from *C. asiatica* showed effective potential in preventing aging [72]. The total triterpenic fractions of *C. asiatica* (TTFCA) have been found to be effective in the management of venous insufficiency and related symptoms, such as ankle edema, foot swelling and capillary filtration rate [73].

The plant possesses good antimicrobial properties. Ethanol and petroleum ether extracts of the *C. asiatica* plant have exhibited significant activity against several fungal strains such as *Aspergillus niger*, *Aspergillus flavus* and *Candida albicans* [64]. Methanol,

chloroform and acetone extracts of *C. asiatica* have been found to exhibit a notable inhibitory effect on the growth and sporulation of *Colletotrichum gloeosporioides* [74]. The ethanol extract also exhibited prominent activity against *Aspergillus niger* and *Bacillus subtilis*, followed by methanol and water [56]. Essential oil extracted from *C. asiatica* exhibited antibacterial activity against Gram-positive (*Bacillus subtilis* and *S. aureus*) and Gram-negative (*Escherichia coli*, *Pseudomonas aeruginosa* and *Shigella sonnei*) bacteria [75].

### 3.3. Nutritional Uses

*C. asiatica* serves as a good source of various macro- and micronutrients (Figure 3). This plant is relatively low in proteins (2.4%), carbohydrates (6.7%) and fat (0.2%). In terms of dietary fibers, it contains insoluble dietary fiber (5.4%) and soluble dietary fiber (0.49%). The mineral contents (mg/g of dry weight) include phosphorus (17.0), iron (14.9) and sodium (107.8) [28–30]. *C. asiatica* is reported to be rich in various vitamins such as vitamin C (48.5 mg), vitamin B1 (0.09 mg), vitamin B2 (0.19 mg), niacin (0.1), carotene (2649 µg) and vitamin A (442 µg) per 100 g approximately [76]. Biochemical experiments show higher total sugar and protein contents in in vitro raised plants than field-grown plants while total starch content was lower in micro-propagated micro-shoots [77].

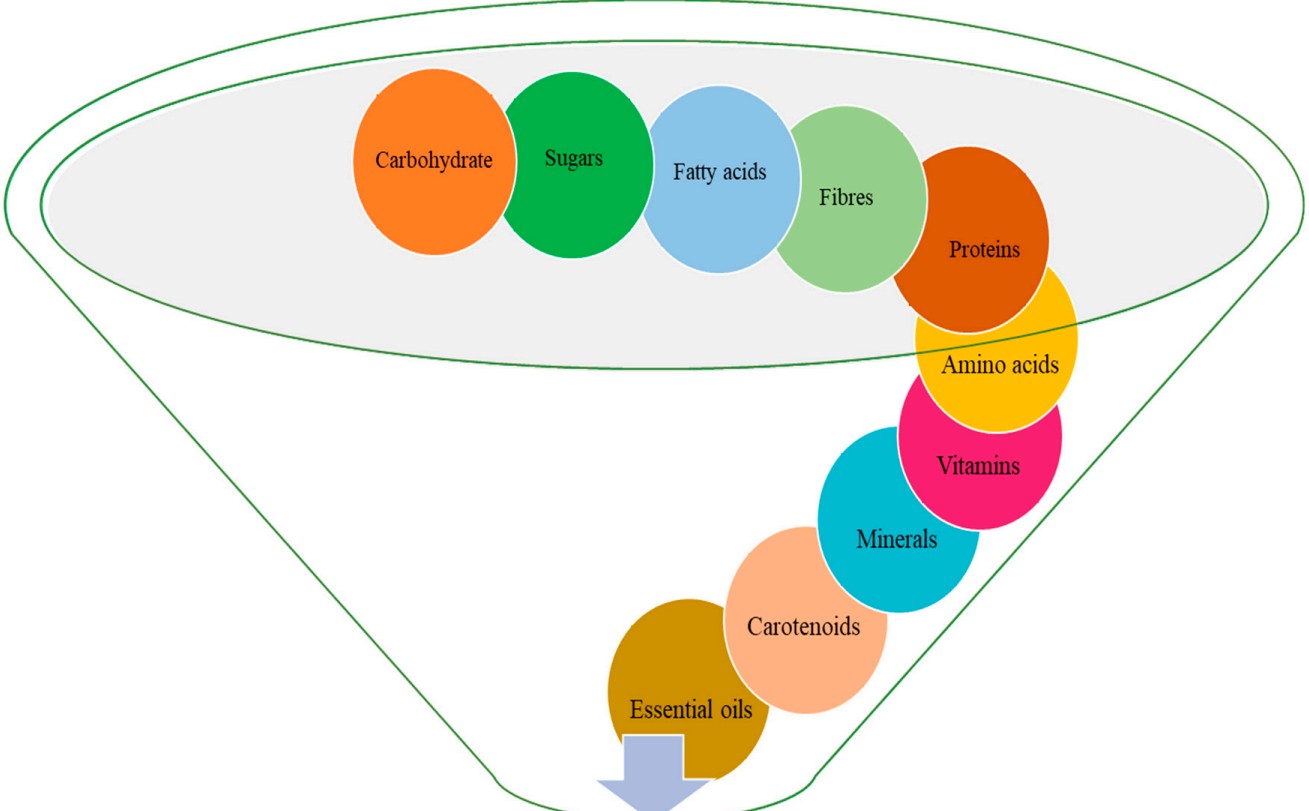

**Figure 3.** Nutritional uses of *C. asiatica*.

### 3.4. Culinary and Nutraceutical Uses

The plant is used as a vegetable in many countries in South East Asia, especially India. In the North East part of India, it holds good economic value as it is sold in open markets and can be grown easily without requiring significant investments. Consuming two to three leaves of *C. asiatica* on an empty stomach in the morning promotes a healthy digestive system and enhances immunity against seasonal and chronic diseases [78,79]. Various ethnic communities in North East India also use it for recreational and medicinal purposes. It is eaten as a salad or cooked as a soup or an appetizer. It is also cooked as a vegetable together with the main meal. Herbal tea is also prepared by steeping either a dried or fresh

plant in boiled water and letting it brew for a few minutes before drinking [80]. It has been used in various polyherbal formulations for commercial applications and health-related purposes. *C. asiatica* is believed to nourish the hair follicles and scalp, reduce hair loss and breakage, strengthen hair strands and promote healthy hair growth. It is used in hair color, anti-dandruff formulations, organic shampoos, hair oils, hair gels, hair conditioners and other hair care products [68]. It is known for its potential to brighten and nourish the skin, reduce pigmentation, fine lines and dark circles and provide overall skin rejuvenation. It is used in skin creams, skin toners, mask packs, cleansing balms, skin moisturizers, facial serums and other skincare formulations (Figure 4) [68].

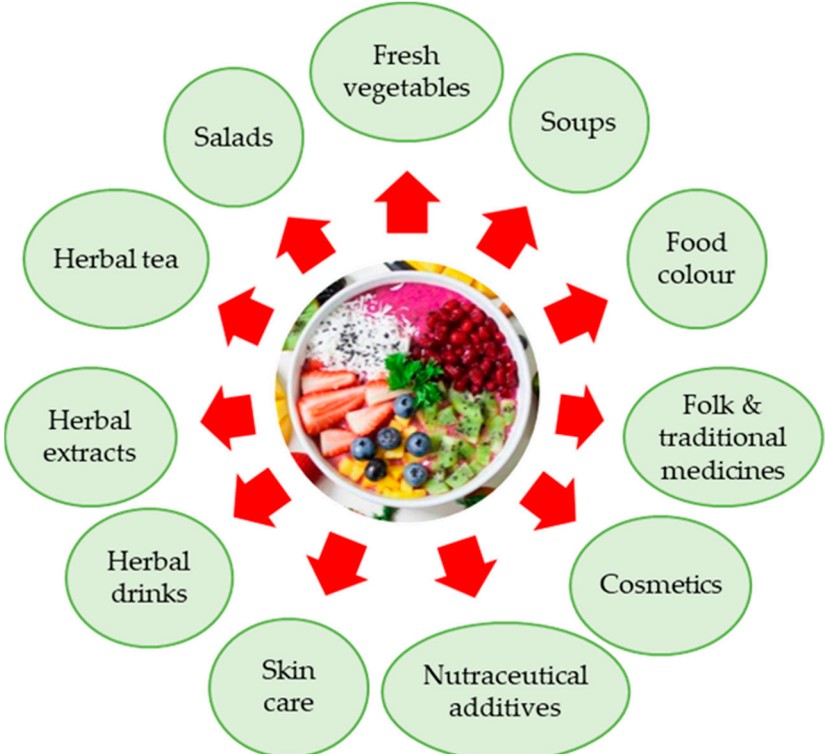

**Figure 4.** Culinary and nutraceutical uses.

*3.5. Major Phytochemicals*

The major secondary metabolites of *C. asitica* are saponin-containing triterpene acids and their sugar esters. Saponins are ubiquitous secondary plant metabolites that are produced through the isoprenoid pathway, which results in a hydrophobic triterpenoid structure (aglycone) with a hydrophilic sugar chain (glycone) [17]. The major phytochemical constituents such as asiatic acid, madecassic acid, asiaticosides, madecassoside, asiaticoside A and asiaticoside B have been characterized [81]. The quantitative estimation of major secondary metabolites in methanol extract by HPTLC indicated the presence of madecassoside, asiaticoside and its sapogenin asiatic acid [82,83]. Madecassoside, asiaticoside, madecassic acid and asiatic acid from *C. asitica* have also been quantitated by HPLC [84]. The content of asiaticoside was 1.20 µg/mL in *C. asiatica* by HPLC [85]. The methanolic extract of the aerial part of the plant resulted in the discovery of three novel compounds, namely centellin, asiaticin and centellicin [86]. The whole-plant essential oil is a colorless mild-scented oil with a yield of 0.06%. The GC-MS spectra revealed forty components, accounting for 99.12% of the oil content. The GC-MS analysis of essential oil also reported p-cymene (44%) as a major constituent [87]. The GC-MS of essential oil chiefly contains sesquiterpenes (68.80%), comprising α-humulene (21.06%), β-caryophyllene (19.08%), bicyclogermacrene (11.22%), germacrene B (6.29%) and germacrene D (4.01%) [88]. The major phytochemicals of *C. asiatica* are presented in Figure 5.

Madecassoside

Asiaticoside

Medacassic acid

Asiatic acid

Isothankunic acid

Quercetin

Rutin

Apigenin

Bayogenin

Betulinic

**Figure 5.** *Cont.*

Brahmol

Terminolic acid

Methyl asiatate

Methyl brahmate

Myricetin

Patuletin

Pomolic acid

Madasiatic acid

Kaempferol

β - caryophyllene

**Figure 5.** *Cont.*

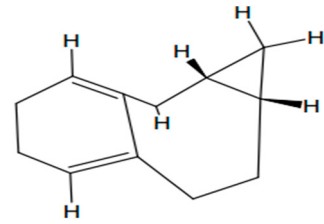

α - humulene

Germacrene B

Bicyclogermacrene

Germacrene D

Centellin

Asiaticin

Castilliferol

Centellasapogenol

p-cymene

**Figure 5.** Major phytochemicals of *C. asiatica*.

### 3.6. Centelloside Content in Accessions

Accessions have been collected for the identification and domestication of superior genotypes for gainful cultivations. Some important research papers on the collection and phytochemical characterization of accession are summarized in Table 1. The selection and propagation of elite genotypes with a higher content of the major centellosides, viz. asiatic acid, madecassic acid, asiaticoside and madecassoside, have been a major objective in *C. asiatica* production in the past few years. It has been reported that the concentration of asiatic acid, madecassic acid, asiaticoside and madecassoside varies between 0.02 and 3.2%, 0.02 and 3.06%, 0.018 and 4.3% and 0.01 and 4.8%, respectively, in various accessions collected from India. The ecological niche modeling approach revealed that the areas with high climatic suitability for the production of *C. asiatica* with a high content of these centellosides are largely confined to the Western Ghats, North East, Eastern Himalaya and Western Himalaya in India [89,90]. The large leaf of *C. asiatica* collected from Varanasi, Uttar Pradesh (central India), showed a low concentration of asiatic acid (0.05%), asiaticoside (0.31%) and madecassoside (0.31%) compared to the above-mentioned agroclimates [80]. Among the five accessions collected from Assam, Meghalaya, Uttar Pradesh, Kerala and Uttaranchal, the samples collected from Assam were found to have the highest content of asiaticoside (21.660 mg/g dry weight), madecassoside (11.176 mg/g dry weight), madecassic acid (0.740 mg/g dry weight) and asiatic acid (1.640 mg/g dry weight) [91]. In accessions collected from south India, the highest content of madecassoside was $5.67 \pm 0.08\%$ (dry weight of the whole plant), while the highest content of asiaticoside was $1.70 \pm 0.02\%$. The combination of asiaticoside and madecassoside in *C.asiatica* collected from Kerala was reported to be $6.18 \pm 0.26\%$ (dry weight of the whole plant) [92].

**Table 1.** Concentration of major phytoconstituents in wild accessions.

| Accessions Collected from Wild | Plant Part Used | Post-Harvest Processing | Concentration of Major Phytoconstituents | References |
|---|---|---|---|---|
| Collected from 2 locations in Varanasi, India | WP (small leaf and large leaf) | SD | Small leaf: asiatic acid (0.04%), asiaticoside (0.34%), madecassoside (0.38%) Large leaf: asiatic acid (0.05%), asiaticoside (0.31%), madecassoside (0.31%) | [83] |
| Collected from specific locations of different phytogeographical zones of India | WP | SD | Asiaticoside (0.66%), madeccasoside (2.688%), asiatic acid (0.44%) | [85] |
| Collected from various agroclimatic zones of India | L | SD | High content of centellosides in Western Ghats of India Madecassoside (4.8%), asiaticoside (4.3%), madecassic acid (3.03%), asiatic acid (2.3%) | [89] |
| Collected from different ecotypic regions of India (Orissa, Uttar Pradesh, West Bengal, Assam and Meghalaya) | L | OD (60 °C for 48 h). | Asiaticoside in dry leaf in germplasm collected from Uttar Pradesh (0.114%), Orrisa (0.062%), Darjeeling, West Bengal (0.097%), Kamakhya Hill, Assam (0.088%), Shillong, Meghalaya (0.105%) | [90] |
| Collected from Eastern, Central and Western Nepal | S | SD | Asiaticoside (0.24 to 8.13%), asiatic acid (0.29 to 0.66%) | [93] |
| Collected from 5 locations (Assam, Uttar Pradesh, Meghalaya, Kerala and Uttaranchal) | S and L | SD | Concentration of centellosides (mg/g dry weight) Highest content from Assam Asiaticoside (21.660), madecassoside (11.176), madecassic (0.740), asiatic acid (1.640) | [91] |
| Collected from vast geographical areas of south India | WP | AD | Highest content of asiaticoside was found in Ooty, Tamil Nadu (1.70 ± 0.02%) Highest content of madecassoside was found in Idukki, Kerala (5.67 ± 0.08%) | [94] |
| Collected from various agroclimatic regions of Kerala. | WP | OD | The highest content was found in Idukki Kerala of asiaticoside and madecassoside combined (6.18 ± 0.26%) | [92] |

WP—whole plant, S—stem, L—leaves, SD—shade-dried, OD—oven-dried, AD—air-dried.

The range of asiaticoside content in *C. asiatica* collected from Central Nepal was reported to be between 0.24% and 8.13%, while the range for asiatic acid content was between 0.29% and 0.66%. The study suggests that plants collected from Central Nepal exhibit higher levels of these metabolites compared to plants from the Eastern and Western regions of Nepal [94]. *C. asiatica* leaves from the local market in Pathumthani Province, Thailand, indicated asiatic acid, madecassic acid, asiaticoside and madecassoside content of 3.39, 4.4, 10.69 and 19.84 mg/g dry weight basis, respectively [95].

*3.7. Production Practices*

3.7.1. Open and Protected Cultivations and Hydroponic Production

The cultivation of *C. asiatica* has been carried out in open air under full sunlight and protected (in glasshouses/shade net houses) with reduced light intensity to enhance productivity and quality. Table 2 summarizes the effect of open and protected cultivation of *C. asiatica* on biomass and phytochemical contents. Accessions collected from Meghalaya, Orissa and West Bengal, India, were cultivated under open air in full sunlight and 50% light in a shade net house. An increase in fresh herb yield (18%) and dry matter yield (41%) was reported in cultivation under 50% sunlight in a shade net house compared to cultivation in full sunlight. The asiaticoside concentration was slightly high in 50% shade (0.91%) compared to open cultivation (0.90%) in full sunlight [96]. Accessions collected from Bhowali, Uttarakhand, India (accessions A), and Bengaluru, Karnataka, India (accession M), were cultivated under open air in Lucknow, Uttar Pradesh, India, and Bengaluru, Karnataka, India. The open cultivation of accession M in Lucknow produced higher fresh herb yield, dry weight yield and asiaticoside, madecassoside, asiatic acid and madecassic acid content compared to cultivation in Bengaluru [85]. This study showed that agroclimate–genotype interactions play a significant role in centelloside content in *C. asiatica*. The asiatic acid content was high in organic cultivation (8.50%) compared to a non-organic cultivation system (7.67%) in an experiment conducted in Nagpur, Maharashtra, India [97]. A significant increase in dry biomass has been observed on the supplementation of inorganic nitrogen fertilizer at 50, 75, 100, 125 kg/ha and 100:60:60 kg/ha of NPK along with a basal dose of FYM (5 t/ha) in Arka Divya and Arka Prabhavi cultivars of *C. asiatica* in open-field and 50%-shade cultivation in Bengaluru, India. The total centelloside contents were higher in open cultivation (4.87%) compared to 50% shade (4.57%) at 50 Kg/ha of nitrogen fertilizer supplementation along with a basal dose of FYM (5 t/ha). The study showed that the open-field cultivation of *C. asiatica* would be more profitable compared to 50% shade due to a better benefit–cost ratio in the selected agroclimatic conditions [98]. A field test involving different lines of *C. asiatica*, specifically the induced tetraploid line and diploid lines, was conducted. The results of this study showed that the tetraploid line exhibited a significant increase in both dry weight and total triterpenoid content compared to the diploid lines [99]. Late harvest of *C. asiatica* is considered the best approach to maximize the production of asiaticoside and madecassoside [100]. The addition of micronutrient copper resulted in a decrease in biomass as well as the concentration of asiaticoside, madecassoside, asiatic acid and madecassic acid in a hydroponic system. The results indicated that the plant has poor copper tolerance in contaminated soils [101]. Favorable environmental and climatic conditions in wetland regions play a significant role in the high concentration of bioactive compounds in *C. asiatica*. Higher contents of phenols, flavonoids and ascorbic acids have been reported in natural environments compared to in vitro propagation [102].

**Table 2.** Effect of production systems on quality attributes.

| Accessions from Wild/Released Variety | Production System/Agronomic Practices | Plant Part and Harvesting Period | Post-Harvest Processing | Quality Attributes | | References |
|---|---|---|---|---|---|---|
| | | | | Yield Parameters | Major Phytoconstituents | |
| Bhowali, Uttarakhand, India | Hydroponic system | L, 42 days | FD | Fresh weight (1.997 ± 0.828 g/2 L hydroponic jar) Dry weight (0.7709 ± 0.206 g/2 L hydroponic jar) | Madecassoside (11.0), asiaticoside (1.7), madecassic acid (36.6), asiatic acid (6.3) mg/g dry weight | [83] |
| Collected from Bhowali Uttarakhand (accession A), and Bengaluru, Karnataka (accession M), regions of India | Open cultivation | WP, After 3 months of planting | FD | Cultivation in Lucknow Fresh weight (t/ha) Accession A (4.687–5.962), M (6.264–7.618) Dry weight (mg/ha) Accession A (1.129–1.237), M (1.197–1.539) Cultivation in Bengaluru Fresh weight (t/ha) Accession A (4.862), M (3.822) Dry weight (mg/ha) Accession A (0.702), M (0.908) | Centelloside concentration (mg/g dry weight) Cultivation in Lucknow Asiaticoside: Accession A (32.7), M (52.1) Madecassoside: Accession A (18.2), M (25.6) Asiatic acid: Accession A (16.7), M (21.7) Madecassic acid: Accession A (44.0), M (9.4) Cultivation in Bengaluru Asiaticoside: Accession A (28.2), M (37.0) Madecassoside: Accession A (63.5), M (63.9) Asiatic acid: Accession A (7.7), M (7.4) Madecassic acid: Accession A (10.6), M (7.8). | [85] |
| Local herbal market of Nagpur, Maharashtra, India | Organic cultivation, open field | L, 1st harvesting | SD | Dry weight (9.82%) | Asiatic acid (8.50%) | [97] |

**Table 2.** *Cont.*

| Accessions from Wild/Released Variety | Production System/Agronomic Practices | Plant Part and Harvesting Period | Post-Harvest Processing | Quality Attributes | | References |
|---|---|---|---|---|---|---|
| | | | | Yield Parameters | Major Phytoconstituents | |
| Released varieties (Arka Divya and Arka Prabhavi), ICAR-IIHR, Bengaluru, India | Cultivation in open field and 50% shade | WP, 2 years | OD (65 °C) | Dry biomass (g/plant) Open field Arka Divya (96.21) Arka Prabhavi (45.61) 50% shade Arka Divya (85.91) Arka Prabhavi (57.58) | Centellosides concentration (kg/ha) Open field Arka Divya (174.1) Arka Prabhavi (108.4) 50% shade Arka Divya(122.8) Arka Prabhavi (98.1) | [98] |
| Collected from several geographical locations of India (Meghalaya, Orissa, West Bengal) | Cultivated under full sunlight and 50% shade | WP, 7 months | Under different levels of shading (85%, 60%, 50% and full sunlight) | Full sunlight Fresh herb yield $(13.38 \times 100$ kg/ha), dry matter yield $(3.90 \times 100$ kg/ha) 50% shade Fresh herb yield $(15.84 \times 100$ kg/ha), dry matter yield $(5.49 \times 100$ kg/ha) | Asiaticoside Full sunlight (0.90%) 50% shade (0.91%) | [96] |
| Malaysian origin | Glasshouse (fringed (F) and smooth leaf (S) phenotypes) | L, 90 days harvest | FD (overnight) | - | Asiaticoside $(1.15 \pm 0.10\%)$ Madecassoside $(1.65 \pm 0.01\%)$ | [103] |
| Paya Rumput, Melaka, Malasiya | Open cultivation | WP, 60 days | OD (40 °C), 4–5 days | Fresh plant $(27.20 \pm 0.14$ t/ha) | Madecassic acid (0.11 mg/100 g), asiatic acid (0.04 mg/100 g), asiaticoside (1 mg/100 g), madecassoside(1 mg/100 g) | [104] |
| Collected from different habitats in Nepal | (a) Open grassland (b) Partially shaded grassland (c) Open agricultural land | WP | SD | - | Open agricultural land Asiaticoside (1.91%), asiatic acid (0.13%) Partially shaded grassland Asiaticoside (1.41%), asiatic acid (0.08%) Open grassland Asiaticoside (1.71%), asiatic acid (0.07%) | [105] |

Table 2. *Cont.*

| Accessions from Wild/Released Variety | Production System/Agronomic Practices | Plant Part and Harvesting Period | Post-Harvest Processing | Quality Attributes | | References |
|---|---|---|---|---|---|---|
| | | | | Yield Parameters | Major Phytoconstituents | |
| Stem cuttings, Serdang, Selangor Malaysia | Glasshouse using $CT_{50}$ + $NPK_{50}$ | WP, 1 year | CO (45 °C for 48 hr) | Fresh herb (171.36 g/10 plants) Dry herb (32.97 g/10 plants) | Asiaticoside (23.73), madecassic acid (11.55), asiatic acid (12.58) ppm/g dw | [106] |
| Stock plants, Ruhlemann's Krauter and Duftpflanzen, Horstedt, Germany | Hydroponic in greenhouse (fertigated with nutrient solutions at 0, 30, 60, 100 or 150% of the N, P or K using Hoagland nutrient solution) | WP, 8 weeks | FD | Highest biomass (g/10 plants dry weight) Nitrogen 60% Whole plant (13.95) Leaf (6.1 g/10) Phosphorus 150% Whole plant (11.46) Leaf (4.56) Potassium 30% Whole plant (19.51) Leaf (8.14 g/10) | Centelloside concentration (mg/g leaf dry weight) It is highest in nutrient solution containing N 0% (87.41), P 0% (78.03) K 0% (78.11) | [107] |
| Deli Serdang, North Sumatra, Indonesia | Open cultivation with application of phosphorus fertilizer (10,20, 30, 40, 50 kg/ha) | L and R, 84 days after planting | - | - | Centelloside (mg/200 mg) Phosphorus 50 kg/ha Leaf asiaticoside (146.09) Leaf asiatic acid (3768.48) Root asiatic acid (91.04) Phosphorus 40 kg/ha Leaf madecassoside (382.40) Root madecassoside(1080.62) Root asiaticoside (162.98) | [108] |
| Bogor, West Java | Open cultivation with addition of 20 t/ha of chicken manure | WP, 3 months | - | - | Asiaticoside (0.34%) | [109] |

Table 2. *Cont.*

| Accessions from Wild/Released Variety | Production System/Agronomic Practices | Plant Part and Harvesting Period | Post-Harvest Processing | Quality Attributes | | References |
|---|---|---|---|---|---|---|
| | | | | Yield Parameters | Major Phytoconstituents | |
| Tetraploid (T1) was derived from a diploid line of Sirirukhachati Nature Park while the 3 diploid D1 Sirirukhachati Nature Park, Salaya campus, Mahidol University, D2 and D3 from local high-quality lines from Mahasarakham province, Bangkok, Thailand | Open cultivation | WP, 4 months | OD (50 °C) | Dry weight (g/m$^2$) Tetraploid (T1): 77.53 $\pm$ 3.07 Diploid (D1): 24.83 $\pm$ 4.02 Diploid (D2): 70.68 $\pm$ 7.14 Diploid (D3): 57.71 $\pm$ 2.82 | Madecassoside (%) Tetraploid (T1): 8.80 $\pm$ 0.60 Diploid (D1): 4.52 $\pm$ 0.53 Diploid (D2): 7.53 $\pm$ 0.23 Diploid (D3): 8.12 $\pm$ 0.18 Asiaticoside (%) Tetraploid (T1): 5.09 $\pm$ 0.29 Diploid (D1): 3.86 $\pm$ 0.54 Diploid (D2): 4.11 $\pm$ 0.13 Diploid (D3): 4.37 $\pm$ 0.24 Madecassic acid (%) Tetraploid (T1): 0.95 $\pm$ 0.13 Diploid (D1): 0.33 $\pm$ 0.07 Diploid (D2): 0.68 $\pm$ 0.14 Diploid (D3): 0.58 $\pm$ 0.33 Asiatic acid (%) Tetraploid (T1): 0.54 $\pm$ 0.04 Diploid (D1): 0.32 $\pm$ 0.08 Diploid (D2): 0.49 $\pm$ 0.07 Diploid (D3): 0.40 $\pm$ 0.18 | [99] |
| - | Experimental field, University of Sisingamangaraja XII, Medan, North Sumatra | L, 70 days | OD (50 °C) | Fresh weight (191.206 g/plants) Dry weight (17.149 g/plants) | Asiaticoside (53.55), madecasosside (358.18) asiatic acid (294.56) µ/mL | [100] |
| Bhowali, Uttarakhand, India | Hydroponic system | WP, 42 days | FD | Fresh weight (24.7 g/plant) Dry weight (2.35 g/plant) | Asiaticoside (1.2 $\pm$ 0.2) Madecassoside (45.2 $\pm$ 1.3) Asiatic acid (2.3 $\pm$ 0.2) Madecassic acid (41.2 $\pm$ 1.7) mg/g dry weight | [101] |

WP—whole plant, R—root, L—leave, SD—shade-dried, OD—oven-dried, FD—freeze drying, CO—convection oven.

Considering the demand of the plant, hydroponic production has been attempted. The details of the production of *C. asiatica* in a hydroponic system are summarized in Table 2. The fresh (1.997 ± 0.828 g) and dry weight (0.7709 ± 0.206 g) of leaves was, respectively, reported from a two-liter hydroponic jar after 42 days of cultivation [83]. The results of this study indicated that the 42-day harvesting period significantly reduced crop maturity. The hydroponic system may have a benefit: the cost ratio for the profitable cultivation of *C. asiatica*. A comparison of the centelloside concentration reported from accessions collected from wild, open and shade cultivations and hydroponic production systems was drawn from the published literature. The data showed that the highest concentration range of asiatic acid (0.11–8.50%), madecassic acid (0.11–4.4%), asiaticoside (0.34–5.21%) and madecassoside (1–2.56%) has been reported in open cultivations (Figure 6).

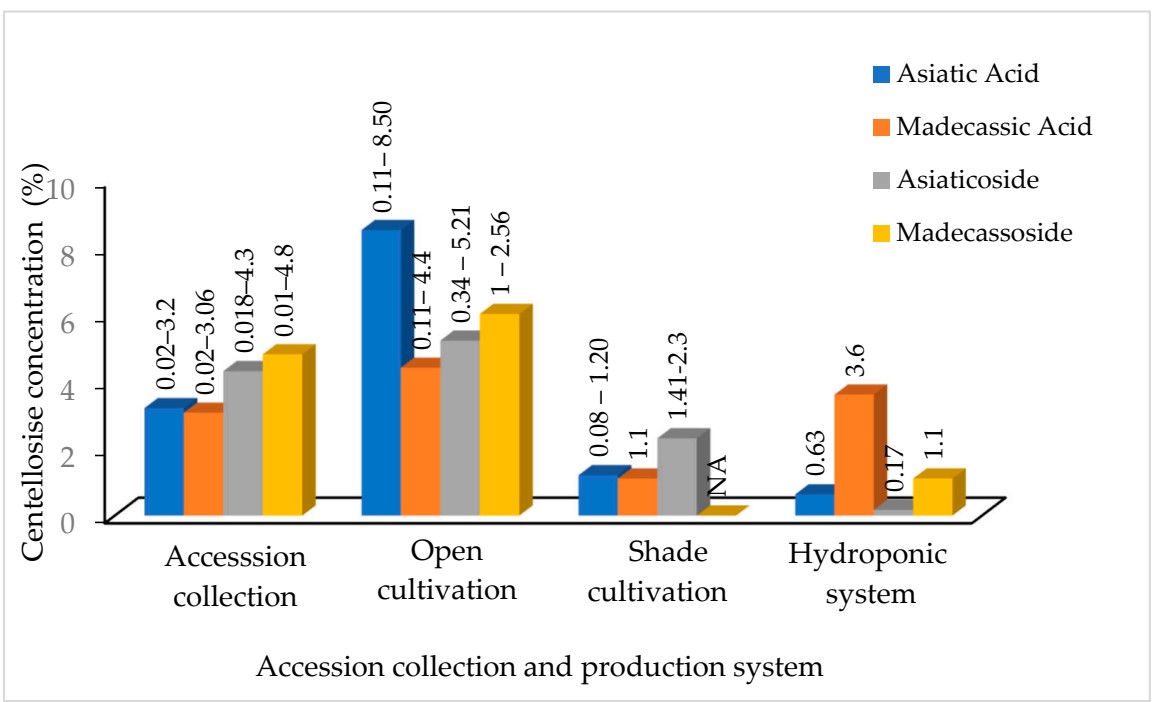

**Figure 6.** Variations in concentration of major centellosides in wild collections and different production systems.

3.7.2. Production in Tissue Culture System

Considering the economic importance of *C. asiatica*, attempts have been made for its production in a tissue culture system. The papers on the tissue culture production of *C. asiatica* are summarized in Table 3. The effects of a number of elicitors, viz. yeast extract, $CdCl_2$, $CuCl_2$ and methyl jasmonate (MJ), on asiaticoside production in the whole-plant cultures of *C. asiatica* were studied. MJ elicited the production of asiaticoside in leaves (9.56 ± 0.91 mg/g dry weight), petioles (1.85 ± 0.07 mg/g dry weight), roots (0.17 ± 0.01 mg/g dry weight) and whole plants (4.32 ± 0.35 mg/g dry weight) in eight-week-old culture [110]. For quality planting material generation, in vitro multiplication resulted in the harvesting of over 25,000 plantlets within 160 days from a single shoot tip explant [111]. The use of 2 mg/L of 2, 4-D and 1 mg/L of kinetin in the establishment of the cell suspension culture of *C. asiatica* proved effective in supporting cell growth and enhancing flavonoid production [112]. When comparing centelloside production in semisolid and liquid media, the production of asiatic acid (1.02 ± 0.03 mg/g fresh weight) in semisolid media was nearly 3-fold compared to liquid media. The leaf callus produced a maximum amount of 1.46 ± 0.06 mg/g fresh weight asiatic acid in semisolid media [113].

**Table 3.** Production of *C. asiatica* in tissue culture system.

| Accessions from Wild | Production Technology | Plant Part and Harvesting Period | Post-Harvest Processing | Quality Attributes | | References |
|---|---|---|---|---|---|---|
| | | | | Yield Parameters | Major Phytoconstituents | |
| Jeju Island, Korea | Tissue culture | WP, 8 weeks in the bioreactor under light culture conditions | FD (24 h) | - | Asiaticoside (mg/g dry weight) Leaves (9.56 ± 0.91), petioles (1.85 ± 0), roots (0.17 ± 0.01), whole plants (4.32 ± 0.35) | [110] |
| - | Tissue culture supplemented with 100 μM methyl jasmonate | C, 15 days | FD | - | Asiaticoside (1.11 mg/g dry weight) Madecassoside (0.62 mg/g dry weight) | [114] |
| - | Tissue culture supplemented with sucrose | C, 24th day of culture | OD (50 °C) | Fresh weight (9.1) Dry weight (1.37) g/50 mL culture | Highest in 30 g/L sucrose Asiaticoside (45.35 mg/g dry weight) | [115] |
| Plants were obtained from Rajiv Gandhi Centre for Biotechnology, Thiruvananthapuram, Kerala, India | Tissue culture colonized with *Piriformospora indica* | WP and L, 45 days | OD | Fresh weight (29.4 g) Dry weight (2.098 g) | Asiaticoside (%) Leaves (0.53) Whole plant (0.23) | [116] |
| - | Tissue culture supplemented with 3% culture filtrate of *Trichoderma harzianum* | Sh, 10th day of 35-day culture | First at room temperature and then at 70 °C in OD | - | Asiaticoside (9.63 mg/g dry weight) | [117] |
| - | Tissue culture supplemented with 2 mg/L 2,4-D, 1 mg/L kintein and 30 g/L sucrose | C, 24 days | OD (50 °C) | Fresh weight (302.45) Dry weight (31.45) g/50 mL of liquid MS media | Asiaticoside (56.21 mg/g) | [115] |
| Collected from National Bureau of Plant Genetic Resources, New Delhi, India | Tissue culture, supplemented with 1 mg/L BAP and 1.5 mg/L naphthalene acetic acid (NAA) elicited with methyl jasmonate | Sh and C, 6 weeks | AD | - | Asiaticoside (mg/mL) Shoot (0.1434 ± 0.004), callus (0.2004 ± 0.0023) Cell suspension culture (0.016 ± 0.0001) Asiatic acid (mg/mL) Shoot (0.2681 ± 0.010) Callus (0.0656 ± 0.0026) | [118] |

**Table 3.** *Cont.*

| Accessions from Wild | Production Technology | Plant Part and Harvesting Period | Post-Harvest Processing | Quality Attributes | | References |
|---|---|---|---|---|---|---|
| | | | | Yield Parameters | Major Phytoconstituents | |
| Diploids and tetraploids of *C. asiatica* plantlets prepared from mother plant obtained from Siriruckhachati Medicinal Plant Garden, Mahidol University, Thailand | Tissue culture supplemented with 3.0% sucrose and containing 5.5 g/L Agar gel elicited with methyl jasmonate | L and P, 20–28 days | OD (40 °C for 48 h) | - | Diploid hairy root culture Asiaticoside (25.87), Madecassic acid (0.79) Asiatic acid (2.24) µg/mg dry weight Tetraploid hairy root culture Asiaticoside (14.39) Madecassic acid (1.85) Asiatic acid (5.69) µg/mg dry weight | [119] |
| Chinhat forest, Lucknow, India | Tissue culture | L, Sh, 35-day-old Sh | FD | Fresh weight (0.2–0.3 g/10 mL culture) | Asiaticoside (3.80 $\pm$ 0.28 mg/g dry weight) | [120] |
| Collected from the agricultural field of Alwarkurichi Village, Tirunelveli District, Tamil Nadu, India | Tissue culture supplemented with auxin and cytokinin | L, R and AR, 60 days | OD (35 °C) | - | Asiaticoside (mg/g) Leaf (16.72), roots (6.42), adventitious root (11.42) | [121] |
| Collected from Rajdhani Agro Farms, Hyderabad | Tissue culture in liquid as well as semisolid MS media | Sh and L, 32 days | - | - | Asiatic acid (mg/g fresh weight) Shoots: semisolid (1.02 $\pm$ 0.03), liquid (0.47 $\pm$ 0.08) Leaves: semisolid (1.46 $\pm$ 0.06) | [113] |
| Collected from Jawaharlal Nehru Tropical Botanic Garden and Research Institute, Thiruvananthapuram, Kerala, India | Tissue culture with colonizing of *Piriformospora indica* | WP, 45 days | OD (40 °C) | Dry weight (2.21 g/60 plants) | Asiaticoside (0.63 mg/g) | [122] |
| - | Tissue culture supplemented with sucrose | WP | - | Dry weight (27.4 g/L) | Asiaticoside (4.26), Madecassoside (2.34) Madecassic acid (0.71), asiatic acid (1.4) mg/g dry weight | [123] |

**Table 3.** *Cont.*

| Accessions from Wild | Production Technology | Plant Part and Harvesting Period | Post-Harvest Processing | Quality Attributes | | References |
|---|---|---|---|---|---|---|
| | | | | Yield Parameters | Major Phytoconstituents | |
| Collected from Tezpur University Campus, Assam, India | Tissue culture | Suspension-cultured cells, 10 days | OD (37 °C) | | Asiaticoside (494.62 mg/g dry matter) | [124] |
| Collected from Peninsular Malaysia | Tissue culture supplemented with 2,4-D: kinetin (2:1 mg/L) | L, 12 days | OD | Fresh weight (0.67 ± 0.02 g/culture) Dry weight (0.041 ± 0.004 g/culture) | Flavanoid (10.75 ± 0.30) Quercetin (0.146 ± 0.003) Luteolin (0.141 ± 0.003) mg/g dry weight | [112] |
| - | Tissue culture supplemented with sodium alginate | L | FD | - | Asiaticoside (0.09 and 0.08%), madecassoside (0.05 and 0.6%), asiatic acid (0.52 and 0.62%), madecassic acid (0.24 and 0.31%) | [125] |
| Namakkal, Tamil Nadu, India | Tissue culture | L | FD | - | Phenols (10 ± 0.3), flavonoids (45) Ascorbic acid (35) mg/g dry weight | [102] |
| Collected from greenhouse of the Garden of Medicinal Plants, Jagiellonian University, Medical College, Kraków, Poland | Tissue culture supplemented with 50 μM MeJa | SC, 6 days after supplementation | - | - | Centellosides (mg/100 g dry weight) Asiaticoside (7.91) Madecassoside (3.81) Phenolic acids (mg/100 g dry weight) Rosmarinic (231–270) Chlorogenic (59–81) Rutin (212–214 mg/100 g dry weight) | [126] |
| - | Tissue culture | HR (P), 6 weeks | - | - | Triterpenoid (46.57 mg/g dry weight) | [127] |

WP—whole plant, Sh—shoots, L—leaves, P—petiole, AR—adventitious root, HR—hairy root, SC—shoot cultures, C—callus, AD—air-dried, FD—freeze drying, OD—oven-dried.

Tissue culture supplemented with 100 µM of MJ produced asiaticoside (1.11 mg/g dry weight) and madecassoside (0.62 mg/g dry weight) in 15-day-old whole plants [114]. Similarly, tissue culture of *C. asiatica* supplemented with sucrose produced fresh plant weight (9.1 g/50 mL culture) and dry weight (1.37 g/50 mL culture) on the 24th day of harvest. It indicated that *C. asiatica* cells were able to accumulate a significant amount of biomass under the influence of sucrose. The highest dry cell weight of 27.4 g/L and the elicited concentration of asiaticoside (4.26 mg/g dry weight), madecassoside (2.34 mg/g dry weight), madecassic acid (0.71 mg/g dry weight) and asiatic acid (1.4 mg/g dry weight) were observed under sucrose supplementation [123]. Asiaticoside (45.35 mg/g dry weight) was very promising on the 24th day of culture at 30 g/L sucrose supplementation [115].

Tissue culture of *C. asiatica* colonized with root endophytic fungus, *Piriformospora indica*, induced asiaticoside production in leaves (5.3 mg/g dry weight) and the whole plant (2.3 mg/g dry weight) in 45 days of culture [116]. *C. asiatica* supplemented with a 3% culture filtrate of *Trichoderma harzianum* produced 9.63 mg/g of asiaticoside on a dry weight basis on the 10th day of culture [117].

Tissue culture supplemented with the auxin and cytokinin growth hormones produced the highest asiaticoside in leaves (16.72 mg/g) compared to roots (6.42 mg/g) and adventitious roots (11.42 mg/g) on the 60th day of culture [121]. Supplementation of 2 mg/L 2,4-dichlorophenoxyacetic acid, 1 mg/L kinetin and 30 g/L sucrose resulted in a 4.5-fold increase in the concentration of asiaticoside (45.35 mg/g dry weight) compared to planta leaves [115]. Tissue culture, supplemented with 1 mg/L BAP and 1.5 mg/L naphthalene acetic acid (NAA) elicited with MJ, produced $0.1434 \pm 0.004$ mg/mL, $0.2004 \pm 0.0023$ mg/mL and $0.016 \pm 0.0001$ mg/mL asiaticoside in shoot, callus and cell suspension culture, respectively, and $0.2681 \pm 0.010$ mg/mL and $0.0656 \pm 0.0026$ mg/mL asiatic acid in shoot and callus culture, respectively [118]. Supplementation of diploid and tetraploid cultures of *C. asiatica* with MJ (50 and 100 µM concentration) induced centelloside production in hairy root culture. The 20–28-day diploid and tetraploid hairy root culture produced asiaticoside (25.87 and 14.39 µg/mg dry weight), madecassic acid (0.79 and 1.85 µg/mg dry weight) and asiatic acid (2.24 and 5.69 µg/mg dry weight), respectively, at a 50 µM concentration [119]. Similarly, 50 µM MJ demonstrated the highest production increase in centellosides, phenols and flavonoids in the shoot cultures of *C. asiatica* [126]. For shoot induction, Murashige and Skoog (MS) media supplemented with 0.5 mg/L 6-Benzylaminopurine (BAP) and 0.1 mg/L NAA resulted in an $82 \pm 2.2\%$ success rate [128], whereas MS media supplemented with 4 mg/L BAP and NAA in combination with 2 mg/L 2,4-D resulted in a 92% success rate [125]. High frequencies of multiple shoot regeneration were reported for shoot tip explants cultured in MS media supplemented with 4.0 mg/L BAP and 0.1 mg/L NAA. Compared to an average of $10.2 \pm 0.38$ shoots per explant [127], 2.0 mg/L BAP supplementation resulted in 15.24 nodes per shoot after 30 days of culture [129]. Multiple shoots were obtained on the supplementation of MS media with 1.0 mg/L BAP and 0.4 mg/L NAA. Additionally, this study reported that profuse and healthy rooting was achieved when using an MS medium supplemented with 0.2 mg/L Indole-3-butyric acid [130].

Comparing hairy root and adventitious root lines, the petiole-derived lines of hairy roots demonstrated the highest growth index, with a value of 37.8. Additionally, the petiole-derived hairy root lines also exhibited the highest concentration of triterpenoids, measuring at 46.57 mg/g [127,131]. In the suspension-cultured cells of *C. asiatica*, the concentration of asiaticoside was measured at 494.62 mg/g of dry weight [124]. The relative concentrations reported (0.09% and 0.08% for asiaticoside, 0.05% and 0.6% for madecassoside, 0.52% and 0.62% for asiatic acid and 0.24% and 0.31% for madecassic acid) suggest that the encapsulation of the plants in alginate did not have a significant impact on the production of these centellosides [125].

*C. asiatica* cell suspension culture 5-L bioreactor produced asiaticoside (60.08 mg/g dry weight) when culture was initiated with an inoculum size of 50 g [132].

*3.8. Methods of Post-Harvest Processing*

The papers on the post-harvest processing of *C. asiatica* plants are summarized in Table 4. Drying in a hot-air oven (40–60 °C) has been the most common method of post-harvest processing. The shade-drying method has also been practiced to reduce the cost [90,110]. For the drying of *C. asiatica* plants produced in tissue culture, freeze drying has been largely practiced due to the low volume of the material [110,114,120]. There are reports that oven drying (OD) and solar drying (SD) have resulted in a decrease in antioxidant activity as well as total centelloside content in comparison to shade drying (SHD), microwave drying (MD) and freeze drying (FD). Freeze drying has been identified as the most efficient method as the bioactive content, antioxidant activity and color were higher in comparison to SD, SHD, OD and MD [133]. The presence of a high concentration of flavonoids in *C. asiatica* leaf, root and petiole has been reported for FD [134]. Microwave blanching in a heat-pump-assisted dehumidified dryer at 40 °C shows the highest total phenolic compound content of about $4.7 \pm 0.08$ mg/g. Additionally, the microwave blanching and heat pump drying (HPD) methods offer the advantage of reduced drying time compared to conventional drying methods [135]. Mixed-mode natural-convection SD was used for drying a bulk quantity of *C. asiatica*. The dryer was capable of handling an initial moisture content of 80% and drying it completely within 4 h. Furthermore, the solar tunnel dryer provided protection against external elements such as rain, insects and dust, ensuring the integrity and cleanliness of the material [136]. The use of SD was effective in reducing the moisture content of *C. asiatica* from 88.3% to 15.9% within a 12 h period. The average temperature inside the dryer was reported to be 45.4 °C, while the relative humidity was measured at 25.8%. The results suggested that the solar dryer was suitable for drying the plant due to the combination of low drying air temperature and a high moisture evaporation rate [137]. The solar-assisted dehumidification system achieved maximum values for pick-up efficiency ($\eta$P) at 70%, solar fraction (SF) at 97% and the coefficient of performance (COP) at 0.3. These values were obtained when drying *C. asiatica*, with an initial wet basis moisture content of 88% and a final moisture content of 15%, using an air velocity of 3.25 m/s. The results suggest that the solar-assisted dehumidification system is suitable for drying heat-sensitive products like *C. asiatica* due to the fact that the drying process is conducted at low air temperature and low relative humidity [138].

The antioxidant activity in *C. asiatica* leaves was higher in freeze drying in comparison with oven drying and dehydration drying [139]. Although freeze drying may be a slow and cost-intensive method, it produced best-quality material for pharmacological activities.

**Table 4.** Effect of post-harvest drying methods on quality attributes.

| Accessions from Wild | Plant Part Used | Post-Harvest Processing | Quality Attributes | | References |
|---|---|---|---|---|---|
| | | | Yield Parameters | Major Phytoconstituents | |
| Collected from a local farmers market, Pathumthani Province, Thailand | L | LPSSD (50 °C) | Fresh leaves (632.45 ± 2.54%) | Asiaticoside (10.96), madecassoside (20.35) Asiatic acid (6.95 ± 0.15) Madecassic acid (8.91 ± 0.70) m mol/g dry weight | [95] |
| Collected from Bari, District- Jajpur, Odisha, India. | L | FD | Dry weight (4.58 ± 0.8%) | Leaves Asiaticoside (36.2 ± 0.11) Madecassoside (4.40 ± 0.09) Madecassic acid (7.7 ± 0.1) Asiatic acid (31.63 ± 0.05) mg/g | [133] |
| Collected from Malaysian Agriculture Research and Development Institute, Selangor, Malaysia | L, R and P | FD | - | Naringin (μg/100 g) Leaves (1204.51 ± 69.8), root (1664.53 ± 79.5), petiole (380.82 ± 25.1) Rutin (μg/100 g) Leaves (332.82 ± 61.8), roots (556.07 ± 56.6), petiole (292.54 ± 10.6) Quercetin (μg/100 g) Leaves (432.65 ± 4.53), roots (505.26 ± 15.4), petiole (340.56 ± 45) Catechin (μg/100 g) Leaves (319.2 ± 26.5), roots (279.05 ± 8.34), petiole (150.3 ± 10) Luteolin (μg/100 g) Leaves (91.66 ± 0.99), roots (not detected), petiole (57.63 ± 6.2) Keampferol (μg/100 g) Leaves (not detected), roots (210.84 ± 8.9), petiole (not detected) Apigeni (μg/100 g) Leaves (109.04 ± 12.1), roots (not detected), petiole (103.79 ± 8.6) | [134] |
| Collected from Khon Kaen province, Thailand | L | Microwave blanching in HPAD (40 °C) | - | Phenols (4.7 ± 0.08 mg/g) | [135] |
| Collected from the local market, Indonesia | WP | SD | Fresh weight (88.3%) Dry weight (15.9%) | - | [137] |
| Collected from local market in Kajang, Selangor, Malaysia | WP | SADS | Fresh weight (88%) Dry weight (15%) | - | [138] |
| Collected from local market in Jeli District, Kelantan, Malaysia | WP | FD | - | Antioxidant (93.97 ± 0.45%) | [139] |

WP—whole plant, L—leaves, R—root, P—petiole, SD—shade-dried, FD—freeze drying, LPSSD—low-pressure superheated steam drying, HPAD—heat-pump-assisted dehumidified, SADS—solar-assisted dehumidification system.

## 4. Conclusions

*Centella asiatica* (L.) Urban has attracted the attention of researchers owing to its promising pharmacological activities. This plant is widely referred to as "brain food" due to its neuroprotective and nerve-tonic properties. Owing to its growing demand for therapeutic applications, large-scale collections from wild and natural habitats have been accelerated, which may result in the loss of germplasm. Hence, research has been conducted on the controlled collection of the accession of *C. asiatica* from natural habitats in the wild. These accessions have been accessed for the quantitative evaluation of bioactive compounds. The elite germplasm has been brought under cultivation for generating quality raw material for preparing standardized and effective formulations. This review summarizes the identification of the ecological niche with elite genotype suitable production (open, shade cultivation, hydroponic and tissue culture) and a post-harvest processing system for high biomass and bioactive compound concentration. The published research indicated that the Indian subcontinent has high climatic suitability for *C. asiatica*, and genotypes with a high content of centelloside were restricted to the Western Ghats, North East, Eastern Himalaya and Western Himalaya in India. Open cultivation of *C. asiatica* is a suitable and cost-effective method for the large-scale production of the plant material. Hydroponic and tissue culture of *C. asiatica* has also been successfully established for the enhanced production of centelloside using supplements and elicitors such as sucrose, auxins, cytokinins, kinetin, methyl jasmonate, etc. Shade drying and freeze drying have been identified as the most efficient methods for the high pharmacological activities of *C. asiatica* extracts.

**Author Contributions:** R.S. and F.I. conceived the general idea of the manuscript. R.S., P.S., U.K.S. and P.P. developed the concept. R.S. and F.I. wrote the manuscript with the help of B.K. and P.K.S. All authors have read and agreed to the published version of the manuscript.

**Funding:** This research was supported by a project grant (Code 6PFE) under the scheme "Increasing the impact of excellence research on the capacity for innovation and technology transfer within USV Timișoara", The APC was funded by University of Life Sciences "King Mihai I", Romania.

**Data Availability Statement:** No new data were created or analyzed in this study. Data sharing is not applicable to this article.

**Conflicts of Interest:** The authors declare no conflict of interest.

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
