# Peer review of "The Effect of Production and Post-Harvest Processing Practices on Quality Attributes in Centella asiatica (L.) Urban—A Review"

_agronomy, doi:10.3390/agronomy13081999_

Round 1
Reviewer 1 Report
Dear Authors
I would like to invite you to read your manuscript as a whole once to find out the presentation flaws. Your readers deserve a unified and clear presentation of data. Please try to homogenize the language, numbering, font size, and most importantly your figures.
Regards
-----------------------------------
Supplementary:
This review tries to address the utilization of phytochemicals of Centella asatica and with the regards to the production methods. The author successfully provided a good source for their readers in order to get an overall informative material to start their research. This paper does do not provide any solution to a scientific question and that is because of the kind of the manuscript it is.
The most important this review paper lacking is the weak presentation that it does have, figures are so poor in resolution and tables are long, fellow chart figure was the only good part of their presentation. Figure 2-4 are super primitive and the figure 5 lack the necessary unity with regards to the method of drawing the chemicals’ structure. Tables are not of a great abbreviation on information in your paper,
The conclusion is not great and does not support the weight that manuscript carries, please work on that.
There are so many unnecessary spaces trough the manuscript, please delete them. Reference numbers are doubled, please fix that.

--
Author Response
Dear Reviewer,
Thank you very much for your significant input. We have now revised the manuscript as per your suggestion. All changes have been made in red. The point-wise response is provided in the attached file (Response to reviewer#1).
Thank you once again, and with regards,
Corresponding author
Reviewer 2 Report
The MS is a review of the medicinal properties of Centella asiatic, of which there are many. The plant has much historic and ethnomedicinal use. The MS includes a nice summary of methods used to cultivate and handle the plant post-harvest. The authors do a good job of summarizing these in figures and tables.
The manuscript's strengths are in the tables and figures, as they are clear summaries of the literature. Much of the narrative text could be streamlined to avoid repeating the tables. Overall the writing is understandable and informative, but it needs an editing hand. Review is particularly needed for some sentence structure and list-making. Additionally, formatting inconsistencies throughout the MS need to be addressed. These are additionally found in the tables.
One item that must be addressed within the MS is the proper and medically-correct use of terms to convey what the plant can and can not do. For instance, the authors use the word "cure" to state that the plant cures different diseases and ailments that do not have a known cure. Conveying medically incorrect information to an audience can be dangerous. The authors need to check other areas where they over-extend the benefits of using the plant.
Overall, the MS was an interesting read that summarizes several studies in a fashion that both researchers and growers can use. I have provided line-by-line comments in highlighted bubbles in the attached document.

Overall, the quality of the English language was very clear. However, a review and edits are needed throughout the paper to improve sentence structure and formatting consistency. A thorough check for needed articles (e.g., the and a) would help with readability. Some spelling check is also required. Lastly, the long lists become confusing and can be interpreted in several ways; thus, they should be rewritten or split into similar verb-based lists. Please see the attached document for some specifics. I did not highlight everything, just some more obvious areas that need attention.
Author Response
Dear Reviewer,
Thank you very much for your significant input. We have now revised the manuscript as per your suggestion. All changes have been made in red. The point-wise response is provided in the attached file (Response to reviewer#2).
Thank you once again, and with regards,
Corresponding author

Reviewer 3 Report
I want to ask one question for authors. You analysed a lot of literature sources about Centella asiatica. Is this plant more use as medicinal plant or as food plant ?
Section 3. Results and discussions (lines 95-104). I think the font size in this paragraph is too large. Please check.
Section 3.1 Ethnomedicinal use of Centella asiatica (Lines 142–160). In this paragraph the font size is not the same. Pease check and correct.
Section 3.2 Therapeutic uses in modern medicine (line 202–209, line 228, line 243, line 256). Please remove unnecessary spaces between words.
Figure 2. Please correct note Figure 2. Therapeutic uses of C.asiatica. I think the font is too large.
Section 3.4. Major phytochemicals (line 299). Please remove unnecessary spaces between words.
Section 3.4. Major phytochemicals (line 310) Please correct word eesntial to essential.
Section 3.4. Major phytochemicals (lines 309–310). You mentionated that one of major constituents in essential oils of C.asiatica is p-cymene. Can you insert structural formula of p-cymene in Figure 5 ?
Section 3.5.1 Open and protected cultivations and hydroponic productions (lines 359, 360, 366, 368, 375, 378, 379). Please change abbreviation Ha to ha.
Section 3.5.1 Open and protected cultivations and hydroponic productions (lines 386, 387). Please remove unnecessary spaces.
Section 3.5.1 Open and protected cultivations and hydroponic productions (line 387). Madecassic acid (0.95 ± 0.13 dry weight). Please insert units of measurement.
Section 3.5.1 Open and protected cultivations and hydroponic productions (line 393–394). Please make spaces between numbers and symbol ±.
Section 3.5.1 Open and protected cultivations and hydroponic productions (line 406). Note 02 litres. Please check and correct.
3.5.2 Production in tissue culture system (lines 411–420, 445, 447, 455, 472). Please check and correct spaces between numbers and symbol ±.
Table 3 (line 422). Please check and correct spaces between numbers and symbol ±.
3.4 Methods of post– harvest processing (lines 502–507). Please check and correct spaces between numbers and symbol ±.
Table 4 (line 528). Please check and correct spaces between numbers and symbol ±.
References (lines 555–875). Your literature sources have double numbering. Please correct.
Author Response
Dear Reviewer,
Thank you very much for your significant input. We have now revised the manuscript as per your suggestion. All changes have been made in red. The point-wise response is provided in the attached file (Response to reviewer#3).
Thank you once again, and with regards,
Corresponding author
